# Nutritional Support Reduces Circulating Cytokines in Patients with Heart Failure

**DOI:** 10.3390/nu16111637

**Published:** 2024-05-27

**Authors:** Aura D. Herrera-Martínez, Concepción Muñoz Jiménez, Ana Navas Romo, José López Aguilera, Manuel Crespin Crespin, Bárbara Torrecillas Baena, Antonio Casado-Díaz, María Ángeles Gálvez Moreno, María José Molina Puerta, Aurora Jurado Roger

**Affiliations:** 1Maimonides Institute for Biomedical Research of Cordoba (IMIBIC), Av. Menéndez Pidal s/n, 14004 Córdoba, Spainbarbara.torrecillasbaena@gmail.com (B.T.B.); mariaa.galvez.sspa@juntadeandalucia.es (M.Á.G.M.); 2Endocrinology and Nutrition Service, Reina Sofia University Hospital, 14004 Córdoba, Spain; 3Immunology Service, Reina Sofia University Hospital, 14004 Córdoba, Spain; 4Cardiology Service, Reina Sofia University Hospital, 14004 Córdoba, Spain

**Keywords:** oral supplements, heart failure, cytokines, mortality, outcomes

## Abstract

Increased inflammation is associated with the pathogenesis of heart failure (HF). Increased circulating levels of cytokines have been previously reported and generally associated with worse clinical outcomes. In this context, the modulation of inflammation-related parameters seems to be a reasonable therapeutic option for improving the clinical course of the disease. Based on this, we aimed to compare changes in circulating cytokines when Mediterranean diet alone or in combination with hypercaloric, hyperproteic oral nutritional supplements (ONS), enriched with omega−3 (n−3) polyunsaturated fatty acids were administered to patients with HF. Briefly, patients were randomly assigned to receive Mediterranean Diet (control group) vs. Mediterranean Diet plus ONS (intervention group). We observed increased circulating levels of IL-6, IL-8, MCP-1 and IP-10. MCP-1 and IL-6 were associated with overweight and obesity (*p* = 0.01–0.01–0.04, respectively); IL-6 and IL-8 were positively correlated with fat mass and CRP serum levels (*p* = 0.02–0.04, respectively). Circulating levels of IL-8 significantly decreased in all patients treated with the Mediterranean diet, while IL-6 and IP-10 only significantly decreased in patients that received plus ONS. In the univariate analysis, MCP-1 and its combination with IL-6 were associated with increased mortality (*p* = 0.02), while the multivariate analysis confirmed that MCP-1 was an independent factor for mortality (OR 1.01, 95%ci 1.01–1.02). In conclusion, nutritional support using hypercaloric, hyperproteic, n-3 enriched ONS in combination with Mediterranean Diet was associated with decreased circulating levels of some cytokines and could represent an interesting step for improving heart functionality of patients with HF.

## 1. Introduction

Heart failure (HF) patients frequently present with cardiac and non-cardiac comorbidities, including obesity, chronic kidney disease, diabetes mellitus, hypertension, atrial fibrillation and sarcopenia [1,2]; specifically, about 50% have more than five non-cardiac comorbidities [3]. All these comorbidities are associated with worse clinical course and increased mortality [4,5].

Among them, sarcopenia has gained special interest, since it severely affects the clinical course of the disease [6]. Specifically, reduced skeletal muscle mass is accompanied by muscle atrophy and decreased quality of muscle tissue; these changes are probably due to the replacement of muscle fibers by fibrotic and adipose tissue, which result in increased fragility and impaired muscle function [7]. Based on this, sarcopenia is associated with increased comorbidity and poorer clinical outcomes, including worse physical performance, higher oxygen consumption, increased hospitalization rates, decreased ventricular function, and lower survival [8,9,10,11].

The incidence of both, sarcopenia and HF, has dramatically increased in the last years, one of the principal causes is the availability of improved therapeutic options for several diseases and the increase in life expectancy [10]. In this context, aging is correlated with increased dysfunctional systemic levels of inflammatory molecules, which cause a chronic low-grade inflammation and consequently tissue degeneration [12]. Several stimuli, such as damage-associated molecular patterns (DAMPs), trigger the nuclear factor kappa-light-chain-enhancer of activated B cells (NF-κB) pathway, and downstream drive the transcription of inflammatory molecules, TNF-α, IL-6, IL-1, and chemokines, which promote the infiltration of inflammatory cells that deteriorate muscle [13,14]. Thus, the reversal of this chronic inflammation status seems to be a reasonable pathway for study and treatment.

It is well known that certain foods, for example, processed meat, food high in sugar, and fat-rich fatty acids, can increase inflammation [15]. In contrast, supplementation with omega-3 (n-3) polyunsaturated fatty acids (PUFAs), such as alpha-linolenic acid (ALA), eicosapentaenoic acid (EPA), and docosahexaenoic acid (DHA), seems to reduce inflammation [16] and improve muscle strength and function but has limited effect on muscle mass gain [17]. In this context, n-3 PUFA administration has been suggested as a potential therapeutic tool to reduce muscle loss and inflammation. Furthermore, it has been suggested that DHA can improve muscle integrity and function by decreasing proteolysis and inflammation in sarcopenia [18].

Nutritional interventions are easy, cost-effective strategies for improving clinical outcomes in different scenarios; in general, they help to improve the clinical condition of the patient, favoring treatment tolerance, decreasing treatment interruptions and modifying the prognosis [6,7]. Additionally, they can improve self-being, perception and quality of life. Furthermore, supplementation with some nutrients is recommended in some cases for decreasing length of stay or infections; these effects are associated with decreased inflammation, especially in postoperative patients with cancer [11,14]. In this context, it is interesting to evaluate the effect of nutritional interventions in other clinical situations associated with chronic inflammation.


**Aim:**


Since sarcopenia and HF are separately associated with chronic inflammation and malnutrition with or without sarcopenia is highly prevalent in patients with HF, we aimed to evaluate some circulating cytokine levels in patients with a recent admission due to HF. The association of these cytokines with nutritional parameters (anthropometric, instrumental and biochemical measurements) was also performed. Finally, their evolution was compared when patients received a nutritional intervention that consisted of a Mediterranean Diet alone with vitamin D supplementation, or Mediterranean Diet in combination with two hypercaloric, hyperproteic, EPA and DHA-enriched nutritional supplements (ONS) per day and vitamin D supplementation during twenty-four weeks.

## 2. Material and Methods

### 2.1. Patients

This study was approved by the Ethics Committee of the Reina Sofia University Hospital (Cordoba, Spain; reference number 5164 approved on 21 October 2021 and updated on 30 May 2023). It was conducted in accordance with the Declaration of Helsinki and according to national and international guidelines. A prospective open label study was performed, wherein a written informed consent was signed by every individual before inclusion into the study. All patients received information before the inclusion and only if accepted to participate, were included. This cohort was initially studied in an open, randomized, controlled, clinical trial (ClinicalTrials.gov number: NCT05848960) [6], in which patients of both sexes, age >18 years old <85 years old (the primary outcomes of the clinical trial was to evaluate changes in body composition and biochemical parameters after nutritional support).

### 2.2. Nutritional Support

In the clinical trial, patients were randomly assigned by the clinical investigator to receive either Mediterranean Diet alone or Mediterranean Diet plus two hypercaloric, hyperproteic ONS per day, with a 1:1 allocation for twenty-four weeks. The ONS was composed with slow-release carbohydrates, fiber mixture and a combination of n-3 and n-6 fatty acids (specifically n-3: 0.52 g/100 mL; n-6: 0.96/100 mL; EPA and DHA: 385 mg/100 mL). ONS were kindly donated by Vegenat Healthcare^®^; bottles were administered every three weeks to the patients only if adherence >75% was achieved. When included in the study, all patients received general education and advice about nutritional support, Mediterranean diet and physical activity; additionally, patients received oral supplementation with calcifediol (in different doses in order to reach levels of sufficiency, defined with a serum 25OH vitamin D levels >30 ng/dL). Nineteen patients were included in each arm. Five patients died during the study period (four in the control arm and one in the intervention arm).

### 2.3. Nutritional Evaluation

A morphofunctional nutritional evaluation was performed as previously described [11,19,20]. Briefly, physical examination included body composition analysis (bioelectrical bioimpedance, abdominal, arm and calf circumferences), functional tests (up and go test and handgrip strength) and nutritional ultrasound of abdominal adipose tissue and rectus-femoris (RF) muscle of the quadriceps. Specifically, vectorial bioelectrical bioimpedance (BIVA) was performed using a NUTRILAB-Akern impedanciometer; this study reported body composition parameters including fat mass, lean mass, water, bone, phase angle, body cell mass (BCME), extracellular mass (ECME). Nutritional evaluation also included anthropometric parameters (calf, arm and abdominal circumference). For handgrip strength, a Jamar^®^ hydraulic dynamometer was used. Nutritional ultrasound was performed using a GE Logiq E9 ultrasound machine and a linear 9L-D probe; specifically, two ultrasounds were performed: (1) abdominal adipose tissue (AT) ultrasound, in which total abdominal adipose tissue, subcutaneous adipose tissue and pre-peritoneal fat were determined, and (2) rectus femoris (RF) ultrasound, in which subcutaneous AT, RF-Y (anteroposterior) and RF-X (transversal) axis, muscle area and circumference were determined. The presence of malnutrition according to the GLIM criteria [21] and sarcopenia (defined as an age and gender adjusted handgrip strength ≤p25) was also determined. Biochemical nutritional analysis was also performed (haemoglobin, lymphocytes, total cholesterol, total, high-density lipoprotein (HDL) cholesterol, low-density lipoprotein (LDL) cholesterol, triglycerides, transferrin, albumin, prealbumin), heart-related markers (N-terminal pro-brain natriuretic peptide (NT-proBNP)) and inflammation markers (C-reactive protein (CRP) and ferritin) were included. Left ventricular ejection fraction (LVEF) measured using transthoracic ultrasound was also evaluated.

### 2.4. Cytokine Measurement

Serum cytokines were quantified by Cytometry Bead Array (CBA, BD Cytometric Bead Array Human Soluble Protein Master, ref. 558264/558265; Becton Dickinson and Company, San Jose, CA, USA). The following cytokines were analysed according to the manufacturer’s instructions: IL-6 (ref. 558276), IL-8 (CXCL8, ref. 558277), IL-10 (ref. 558274), MCP-1 (CCL2, ref. 558287) and IP-10 (CXCL10, ref. 558280). For sample acquisition, a FACS Canto II was used, and a minimum of 300 events were recorded per each cytokine. Median Fluorescence Intensity (MFI) data were transformed in concentration (pg/mL) using a calibration curve as a reference.

### 2.5. Statistical Analysis

The Kolmogorov–Smirnov test was used to assess the normal distribution of data. For the descriptive statistics, the mean and standard deviation of the continuous variables and the frequencies and percentages of the discrete variables were calculated. To assess differences between the continuous variables, the Mann–Whitney U test was used (nonparametric data). Paired analysis was performed by Wilcoxon test (nonparametric data). For differences between the discrete variables, Pearson’s test was used. Statistical analyses were performed using SPSS statistical software version 20, and Graph Pad Prism version 6. Significance was defined as a *p*-value of <0.05.

## 3. Results

### 3.1. Baseline Characteristics of the Groups

Thirty-eight patients were included. Most of them were male (71.10%) with a mean age of 66.71 y old. The mean baseline ejection fraction was 38.50% and patients presented with a mean of NT-pro-BNP of 5768 pg/mL during the previous hospital admission. Only 15.8% of patients presented gastrointestinal symptoms, but body weight loss during the previous three months was observed in 55.26% of them. According to the GLIM criteria, 23.68% of patients presented with malnutrition, while 65.79% presented with sarcopenia according to the handgrip strength. Specific baseline characteristics are depicted in Table 1.

After six months of nutritional intervention, patients increased their body mass index (BMI; from 28.40 ± 4.74 to 29.30 ± 4.51, *p* = 0.02) and tended to increase fat mass (from 21 ± 10 to 28.10 ± 6.71, *p* = 0.07). The mean of other body composition determined by BIVA and nutritional ultrasound remained stable (Table 2).

Regarding functional parameters, despite no significant changes being observed in handgrip strength, the up-and-go test significantly decreased (difference of 9.42 s, *p* < 0.001), revealing improved functionality. When biochemical parameters were analysed, patients increased their haemoglobin levels and decreased triglycerides, but no significant changes were observed in albumin, prealbumin, or transferrin levels. Regarding heart-related parameters, NT-proBNP levels significantly lowered from 3225 ± 3882 pg/mL to 1300 ± 1226 (*p* < 0.01). Concerning inflammation-related parameters, ferritin significantly dropped (mean of 124.02 ± 99.21 to 86.13 ± 77.42 mg/dL, *p* = 0.003), in parallel to CRP, which changed from 8.54 ± 14.98 mg/L at baseline to 2.82 ± 4.81 mg/L after the nutritional intervention (*p* = 0.02). Detailed changes in biochemical parameters are depicted in Table 2.

### 3.2. Clinical Associations between Circulating Cytokine Levels and Heart Failure (HF)

In this cohort, we observed that patients with HF and BMI > 25 Kg/m^2^ presented with increased levels of IL-8, MCP-1 and IL-6 in combination with IL-8 (Figure 1A). Additionally, in a univariate analysis, patients that died during follow up presented with increased levels of MCP-1 and the combination of IL-6 with MCP-1 (Figure 1B). Any other significant association between cytokine levels and clinical characteristics was observed in the univariate analysis.

### 3.3. Relevant Correlations between Circulating Cytokine Levels and Heart Failure (HF)

At baseline, IL-6 presented significant, positive but weak correlations with BMI (r: 0.32), fat mass (r: 0.39 for fat mass in % and r: 0.39 for fat mass in Kg) and serum CRP (0.40). In contrast, it negatively correlated with lean mass (in %, r: −0.38) and serum LDL (r: −0.36). IL-8 not only correlated with BMI (r: 0.45) and fat mass (r: 0.37 for fat mass in % and r: 0.42 for fat mass in Kg), but also with arm circumference (r: 0.39); similar to IL-6, it negatively correlated with lean mass (r: −0.37; Figure 2A). MCP-1 only correlated positively with RF-X axis (r: 0.40), while IP-10 correlated with the up-and-go test (r: 0.36) and serum levels of NT-proBNP during hospital admission (r: 0.39). Circulating cytokines also correlated among them at baseline in patients with HF, specifically, IL-6 correlated with IL-8 (r: 0.66), MCP-1 (r: 0.40) and IP-10 (r: 0.36), while IL-8 positively correlated with MCP-1 (r: 0.46). Only significant baseline correlations are depicted in Figure 2A.

After 24 weeks of nutritional intervention, IL-6 only positively correlated with RF-AT (r: 0.40) and serum CRP (r: 0.52), it negatively correlated with 25-OH vitamin D levels (r: −0.39), while IL-8 showed more significant, positive, but still weak clinical correlations with body weight (r: 0.45), calf circumference (r: 0.36), abdominal circumference (r: 0.39) and RF-area (r: 0.36); it also negatively correlated with HbA1c. IP-10 level correlated positively with RF-AT (r: 0.52), but negatively with HbA1c (r: −0.53) and serum triglycerides (r: −0.46). After 6 months of follow-up, IL-8 only positively correlated with MCP-1 and IP-10 (Figure 2B).

Finally, when absolute change in the clinical variables and cytokine levels are determined, IL-6 only positively correlated with fat mass (r: 0.48) and negatively with lean mass (r: −0.48) and calf circumference (r: −0.47). Moreover, change in serum IL-6 correlated with the absolute change observed in circulating IL-8 levels (Figure 2C). The absolute change in MCP-1 clinically correlated with the six-month change in ECME in a positive manner (r: 0.42) and negatively with serum ferritin (r: −0.46). We did not observe significant correlations between NT-proBNP or LVEF and circulating cytokines.

### 3.4. Clinical Changes in Circulating Cytokines after Nutritional Interventions in Patients with HF

When circulating serum levels of cytokines were analysed in the whole cohort, we observed that patients that received nutritional support during 24 weeks significantly decreased serum levels of IL-8. Moreover, patients receiving the Mediterranean diet in combination with the ONS, also significantly decreased circulating IL-6 and IP-10. Remarkably, MCP-1 did not change after six months of nutritional intervention in patients with HF (Figure 3).

### 3.5. Clinical Association between HF-Related Outcomes and Circulating Interleukins

An age- and sex-adjusted multivariate analysis showed that the only cytokine associated with increased mortality in patients with HF was MCP-1 (OR 1.01, 95% CI: 1.01–1.02), as well as its combination with IL-6 (OR 1.01, 95% CI: 1.01–1.02). In contrast, no circulating cytokine was associated with new hospital admissions due to HF during the 24 weeks of follow-up (Table 3).

## 4. Discussion

The prevalence of HF has increased during the last decades, currently reaching 2% of the population in developed countries. According to the European Society of Cardiology, it is associated with increased morbidity, institutionalization, and mortality [22], especially in elderly people, due to improved life expectancy and therapeutic options [23].

Malnutrition can affect 10–50% of patients, particularly in advanced HF, and has been associated with a worse prognosis, hospital stay and readmission rate [24,25]; furthermore, in some series, malnutrition has been considered an independent predictor of mortality [26].

The consumption of PUFAs has been reported to improve the prognosis of several chronic inflammatory diseases, including atherosclerosis, specifically supplementation with EPA and DHA, which are highly polyunsaturated and easily undergo auto-oxidation. This oxidation is necessary for achieving their anti-inflammatory effects; specifically, these oxidized products inhibit the cytokine-induced activation of NF-κB and promote cytosolic retention of the p50 and p65 subunits, producing in consequence the inhibition of other cytokines release, such as IL-8 and MCP-1 [27]. Importantly, their effects are dose-dependent; a daily consumption of 2g/day of EPA and DHA are recommended [28]; even doses of 4.8 g/day have shown anti-inflammatory-related benefits in glucose and lipid metabolism that have been reported in humans in a safe manner [29]. Specifically, it has been proposed that high doses of EPA and DHA can modulate T CD4+ lymphocyte subsets activation, differentiation, and proliferation, which, combined with cytokine secretion modulation, would be responsible for their anti-inflammatory effects [29,30].

In this context, it is well known that inflammation plays a role in the pathophysiology of HF [31]. Recurrent and sustained immune system activation has been implicated in the left ventricular hypertrophy and its progression to HF. Elevated circulating cytokines correlate with the severity of HF and prognosis of the disease [32]. Cytokines can increase the rate of cardiomyocyte apoptosis, cardiac hypertrophy and matrix metalloproteinase activation, and thus may affect heart functionality [33]. Based on this, we focused on the analysis of IL-6, IL8-, MCP-1 and IP-10 in patients with HF with a previous hospital admission due to this reason.

It has been described that IL-6 controls both inflammatory and immune responses [34]. It is secreted by adipocytes, particularly visceral fat adipocytes [35]; it is further responsible for synthesis of acute phase proteins in the liver such as CRP and fibrinogen. In HF patients, elevated IL-6 levels have been associated with an increased risk for developing incident HF [36], especially if cardiorenal syndrome, iron deficiency or anaemia are present [37], affecting in particular patients with preserved LVEF, in which is independently associated with increased risk of death or HF hospitalization [37,38].

A recent report described enhanced IL-6 levels in patients with HF and preserved LVEF; IL-6 was associated with higher BMI, total fat mass, trunk fat mass, as well as with raised serum NT-proBNP, CRP and TNF-α levels; furthermore, it was associated with worse renal function, lower haemoglobin levels, and impaired glucose metabolism [39]. Despite our evaluation of patients with intermediate or reduced LVEF, in our study, IL-6 was positively correlated with BMI, fat mass and CRP, and even after nutritional intervention, it still correlated with RF-AT. As previously described in the literature, IL-6 in combination with IL-8 was elevated in patients with overweight or obesity. Previous studies have reported that increased IL-6 levels are strongly associated with reduced exercise capacity and more severe symptoms of exercise intolerance, which are typically present in patients with preserved LVRF and obesity [39]; in contrast, we did not observe these associations. As observed in our cohort, nutritional interventions can modulate IL-6 levels, specifically, n-3 PUFA can reduce IL-6 levels not only in patients with cancer [40], but also in middle-aged and older adults [41].

Regarding IL-8, its production is induced by several stimuli such as shear stress, ischemia and hypoxia, through the activation of the NF-κB pathway, playing a role in angiogenesis, neutrophil and granulocyte chemotaxis, and phagocytosis stimulation [42]. Regarding cardiovascular diseases, IL-8 has a role in atherogenesis and atherosclerotic plaque destabilization; it has been reported as a risk factor for HF and all-cause mortality in several large observational studies [43], and it has been described as a predictor of HF in patients with previous heart infarction [44]. Nevertheless, concerning cardiovascular complications and mortality, results are contradictory [43]. In our cohort, IL-8 positively correlated with IL-6 and MCP-1 levels, and was associated with overweight, obesity and fat mass, as previously reported in other cohorts [45,46]. Moreover, IL-8 levels decreased after 24 weeks of nutritional intervention; this fact has not been reported previously in humans, but in vitro studies suggest and n-3 fatty acids can reduce IL-8 secretion in cultured cell lines [47].

MCP-1 acts as a chemotactic factor that recruits monocytes into the vascular wall [48]; it is also expressed in human macrophage-rich atherosclerotic plaques [49], and has been positively correlated with different cardiovascular risk factors [50]. An association between MCP-1 levels, obesity, diabetes and the presence of diabetes-related complications have been reported [51]; despite our study including a low proportion of patients with diabetes, we clearly observed associations between MCP-1, overweight and obesity. Raised MCP-1 levels have been observed in patients with stable coronary heart disease and with peripheral artery disease [52]; despite vascular ultrasound not being performed in our study, MCP-1 levels were markedly elevated in this cohort and closely correlated with IL-6, which suggests that this association was also present. Furthermore, elevated MCP-1 plasma levels have been described as a risk factor for myocardial infarction and death in patients with acute coronary syndrome [53], particularly in patients with systemic inflammation [54]. In our cohort, MCP-1 was an independent factor for mortality. In this sense, according to some authors, MCP-1 levels seem to provide prognostic information independently of that provided by CRP in patients with cardiovascular events, and are elevated in parallel to serum NT-proBNP [54].

It has been previously suggested that oxidized n-3 fatty acids could inhibit NF-κB activation via a peroxisome proliferator-activated receptorγ (PPARγ)-dependent pathway [27]. Nevertheless, in our cohort, MCP-1 was the only cytokine that did not change after 24 weeks of nutritional intervention. In this respect, a recent work revealed that MAPKs/AP-1 but not NF-κB signaling could be responsible for MCP-1 production in TNF-α-activated adipocytes [55], which might explain the absent response to a nutritional intervention.

Focused on IP-10 (also called CXCL10), this chemokine is secreted by innate immune cells (including monocytes and neutrophils), but also by endothelial cells and fibroblasts, which are dedicated to respond to cytokines and DAMPs emitted upon tissue injury [56]. This is probably the less described cytokine in HF; increased IP-10 levels have been reported in patients with coronary heart disease [57]. In our cohort, we observed a positive correlation between IP-10 and NT-proBNP serum levels during hospital admission. A recent study suggested that increased serum levels of IP-10 in the acute phase of after ST-segment elevation myocardial infarction can predict a better recovery in cardiac systolic function and less adverse events in these patients [58]. But the translational role of this cytokine in HF prognosis still needs to be determined. The literature about the effect of nutritional intervention on IP-10 is scarce; nevertheless, a reduction on this chemokine production has been described for resveratrol treated M1 activated macrophages [59]. In our cohort, IP-10 circulating levels also decreased in patients that received nutritional support with OS.

## 5. Limitations

This study has some limitations, first the number of participants in each group, which can limit the findings of this study, this fact combined with individual variations make difficult to obtain solid conclusions. Furthermore, we cannot determine a specific relation between one component of the nutritional support and the clinical effect, since we used an ONS and not separate components (e.g., n-3 fatty acids, whey protein modules, etc). Cytokines are not specifically related with sarcopenia and other situations could affect (increase or decrease their levels). HF is a chronic disease associated with increased inflammation, thus, the disease itself can affect circulating cytokine levels. Finally, the underlying molecular mechanism were not evaluated. In contrast, this study has several strengths, including the duration of the study, the confirmation of the adherence to the ONS and the comprehensive nutritional evaluation that was performed. Ultimately, to the best of our knowledge, this is the first report of a nutritional intervention on circulating cytokines in patients with HF.

## 6. Conclusions

Taken together, our results reveal a close relation between circulating cytokines and body composition parameters in patients with FH. Furthermore, a nutritional intervention with the Mediterranean diet results in decreased IL-8 circulating levels, while its combination with an ONS (with slow-release carbohydrates and fiber mixture and enriched with EPA and DHA) resulted in an additional significant decrease in serum IL-6 and IP-10, suggesting that nutritional intervention can affect the clinical evolution of the heart function patients with previous admissions due to HF. Finally, MCP-1 was the only parameter independently associated with mortality and was the only that not significantly changed after the nutritional intervention, emphasizing the potential role of pathways other than NF-KB in the pathogenic mechanisms underlying this cytokine.

## Figures and Tables

**Figure 1 nutrients-16-01637-f001:**
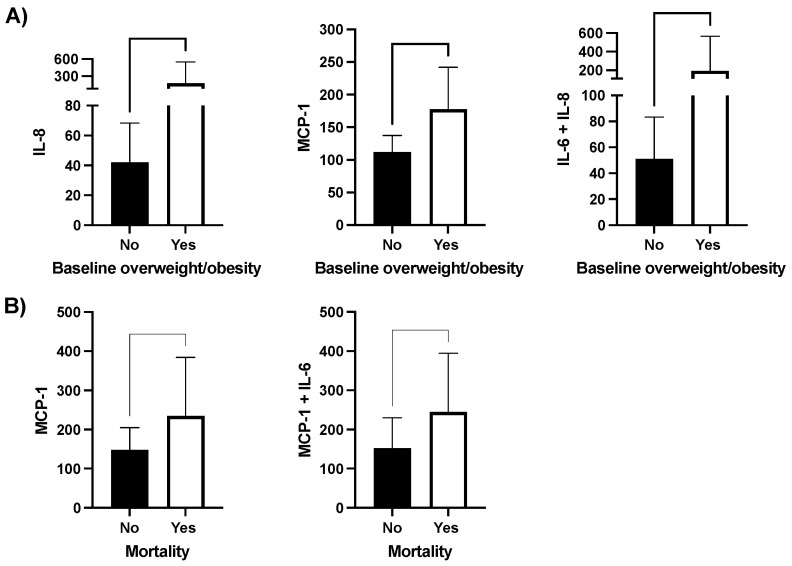
Clinical associations between circulating cytokine levels and clinical characteristics of patients with HF. (**A**) Cytokine levels in patients with baseline overweight or obesity (n = 10) vs. patients with BMI < 25 kg/m^2^ (n = 28). (**B**) Serum cytokine levels in patients with heart failure that survived and died during the study (n = 5 and n = 33 respectively. Legend: only significant associations are depicted.

**Figure 2 nutrients-16-01637-f002:**
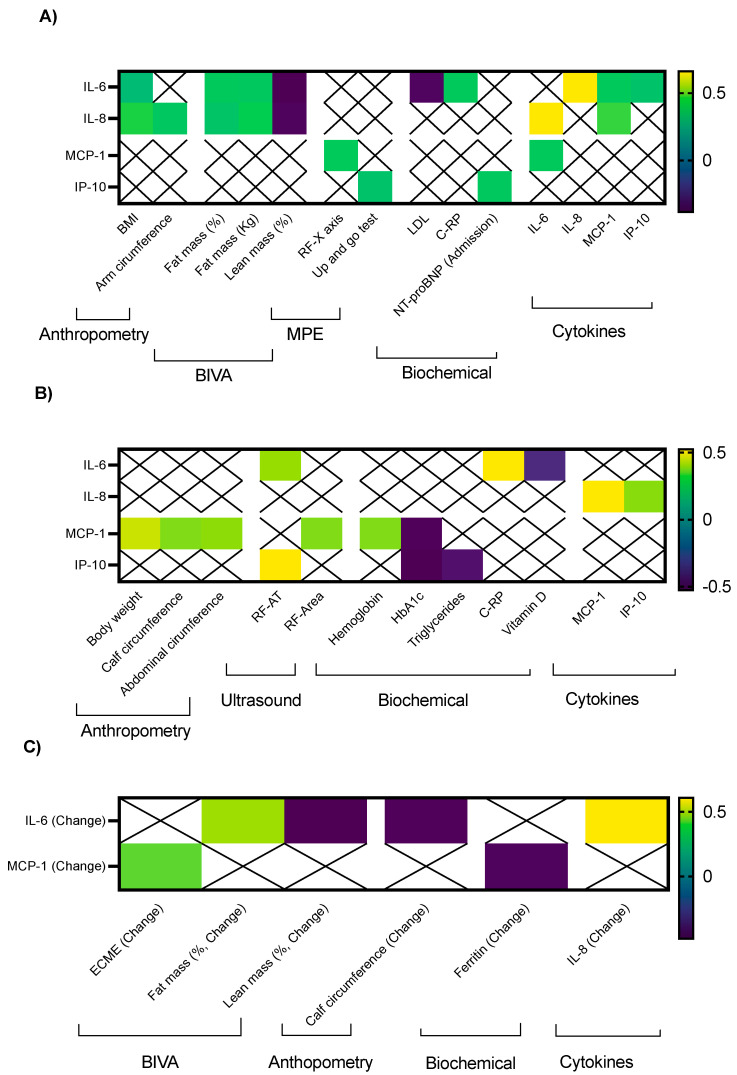
Significant clinical correlations between circulating cytokines, body composition and biochemical parameters in patients with HF: (**A**) clinical correlations observed at baseline; (**B**) clinical correlations determined after 24 weeks of nutritional intervention; (**C**) six months absolute changes. Legend: only significant correlations are presented. Color boxes represent the strength of the correlation according to the color scale represented on the right side of each figure. BIVA: bioimpedance vectorial analysis, MPE: morphofunctional evaluation; RF: rectus femoris; CRP: C reactive protein.

**Figure 3 nutrients-16-01637-f003:**
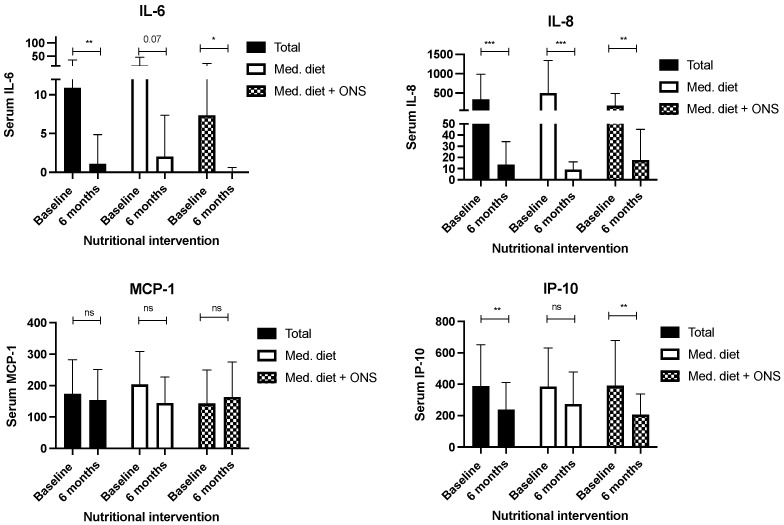
Effect of a 24-week nutritional intervention (based on Mediterranean diet with or without an oral n-3 enriched nutritional supplement) on circulating cytokine serum levels in all the evaluated patients with HF. Legend: ONS: oral nutritional supplement; serum cytokines are measured in pg/mL. * *p* < 0.05; ** *p* < 0.01; *** *p* < 0.001; ns: non-significant.

**Table 1 nutrients-16-01637-t001:** Baseline clinical characteristics of the patients. Comparison between groups based on the nutritional intervention.

Characteristics	Total (n = 38)
Sex (♂/♀)	71.10%/28.90% (11/27)
Age (years)	66.71 ± 13
Tobacco exposure (%)	
No	57.89 (22/38)
Active	18.42 (7/38)
Previous exposure	23.68 (9/38)
Type 2 Diabetes (%)	42.11 (16/38)
Previous ischaemic cardiomyopathy (%)	34.21 (13/38)
Ejection fraction (%)	38.50 ± 16
NT-proBNP (pg/mL)	5768 ± 6646
Current weight (Kg)	68.51 ± 13.5
Symptoms	
Weight loss (3 months, %)	55.26 (21/38)
Weight loss Kg (3 months)	2.43 ± 2.7
Weight loss (6 months, %)	28.94 (11/38)
Weight loss Kg (6 months)	1.72 ± 3
Uncomplete denture (%)	63.16 (24/38)
Food intake (%)	
Soft	7.89 (3/38)
Normal	92.11 (35/38)
Gastrointestinal symptoms (%)	15.79 (6/38)
Abdominal pain	10.53 (4/38)
Nauseas/vomits	5.26 (2/38)
Diarrhea	5.26 (2/38)
Body lesions	0
Dyspnea	78.94 (30/38)
Malnutrition (%)	23.68 (9/38)
Sarcopenia (%)	65.79 (25/38)
Physical activity (%)	
Intense	0
Moderate	18.42 (7/38)
Resting time (hours/day)	8 ± 4
Quality of life	
Self-rated health score	69 ± 22

**Table 2 nutrients-16-01637-t002:** Changes in the morphofunctional and biochemical assessment of the nutritional status at baseline and six months after nutritional intervention.

	Total		
Characteristics	Baseline (n = 38)	Six Months (n = 32)	*p*
Body weight	77.81 ± 16.11	79.62 ± 17.21	0.02
Bioimpedance analysis			
BMI (Kg/m^2^)	28.40 ± 4.74	29.30 ± 4.51	0.02
BCMe	36.31 ± 7.33	36.31 ± 7.41	0.67
ECMe	25.82 ± 4.71	27.11 ± 4.83	0.24
Fat mass (%)	26.20 ± 2.22	27.60 ± 1.81	0.19
Fat mass (Kg)	21 ± 10	28.10 ± 6.71	0.07
Lean mass (%)	70.01 ± 8.80	69.01 ± 8.62	0.35
Lean mass (Kg)	54.21 ± 10.13	54.41 ± 10.32	0.30
Water (%)	52.19 ± 7.04	51.01 ± 6.23	0.43
Water (Kg)	40 ± 8.44	52.9 ± 10.90	0.40
Bone Mass (Kg)	2.89 ± 0.51	3.82 ± 0.81	0.52
Phase angle	5.03 ± 2.32	4.92 ± 1.94	0.80
Anthropometric evaluation			
Abdominal circumference	104.38 ± 12.02	105.26 ± 13.94	0.90
Arm circumference	29.89 ± 3.81	30.10 ± 3.51	0.49
Calf circumference	37 ± 4.80	37.30 ± 3.53	0.53
RF-Muscle Ultrasound			
Adipose tissue (cm)	0.81 ± 0.41	0.78 ± 0.13	0.53
Area (cm^2^)	3.72 ± 2	3.10 ± 1.52	0.19
Circumference (cm)	8.63 ± 2	8.27 ± 1.31	0.42
AP axis (cm)	1.13 ± 0.61	1.08 ± 0.23	0.20
Transversal axis (cm)	3.71 ± 0.91	3.56 ± 0.40	0.42
Abdominal Ultrasound			
Total adipose tissue (cm)	2.33 ± 0.22	2.52 ± 1.11	0.87
Subcutaneous adipose tissue (cm)	2.71 ± 0.12	1.73 ± 0.82	0.53
Preperitoneal fat (cm)	0.71 ± 0.31	0.74 ± 0.12	0.37
Functional evaluation			
Handgrip strenght (dominant arm, Kg)	31.02 ± 11.51	30.81 ± 13.81	0.71
Up and go test (seconds)	21.54 ± 9.0	12.12 ± 4.22	<0.001
Biochemical parameters			
Haemoglobin	13.98 ± 1.8	14.4 ± 1.49	0.03
Lymphocytes	1827 ± 611	96.5 ± 14.46	0.13
Albumin (g/dL)	4.51 ± 0.52	27 ± 5	0.30
Prealbumin (mg/dL)	24.45 ± 6.43	36.52 ± 3.50	0.55
Ferritin (mg/dL)	124.02 ± 99.21	86.13 ± 77.42	<0.01
Transferrin (mg/dL)	254 ± 51.94	223 ± 34.67	0.12
Total cholesterol (mg/dL)	163 ± 46	164 ± 46	0.72
HDL cholesterol (mg/dL)	47 ± 13	52 ± 19	0.17
LDL cholesterol (mg/dL)	87 ± 39	83 ± 21	0.12
Triglycerides (mg/dL)	235.11 ± 52.03	123 ± 55.34	<0.001
CRP (mg/L)	8.54 ± 14.98	2.82 ± 4.81	0.02
NT-proBNP (pg/mL)	3225 ± 3882	1300 ± 1226	<0.01
Vitamin D (ng/dL)	19.50 ± 10.33	23.95 ± 13.12	0.08

Legend: RF: rectus femoris. CRP: C reactive protein *p*1 refers to the comparison between all patients at baseline and after twenty-four weeks.

**Table 3 nutrients-16-01637-t003:** Multivariate logistic regression for mortality and new hospital admissions in patients with HF that received nutritional support after adjusting by age and sex.

Variable		OR	CI	*p*
Mortality	Baseline IL-6	0.98	0.93–1.05	0.66
	Baseline IL-8	1.00	0.90–1.00	0.76
	Baseline MCP-1	1.01	1.01–1.02	0.03
	Baseline IP-10	1.00	0.99–1.00	0.73
	Baseline IL-6 + MCP-1	1.01	1.01–1.02	0.04
	Baseline CRP	0.99	0.94–1.06	0.98
New hospital admissions	Baseline IL-6	1.01	0.98–1.05	0.23
	Baseline IL-8	0.99	0.99–1.00	0.58
	Baseline MCP-1	1.00	0.99–1.01	0.55
	Baseline IP-10	1.00	0.99–1.00	0.66
	Baseline CRP	1.01	0.97–1.06	0.48

## Data Availability

The original contributions presented in the study are included in the article, further inquiries can be directed to the corresponding author.

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
