# Peer review of "Nutritional Support Reduces Circulating Cytokines in Patients with Heart Failure"

_nutrients, 2024, doi:10.3390/nu16111637_

Round 1

Reviewer 1 Report

Comments and Suggestions for Authors

Interesting work, but the manuscript should be improved:

Nutritional support and Circulating cytokines in patients with heart failure. Is it possible to change the clinical course of these patients?

The title opens a question that is expected to be resolved in the text, but I am not sure if they answer it. Maybe the title should be changed.

Line 21... the phrase is in a different letter typo, same in line 45 (and lines 6, 59, 141,142), line 30 C-RP? Maybe is CRP

In methodology, authors should include the n-3 (EPA, DHA) and omega-6 (LA?) concentration in every dose administrated.

19 patients is a low average for doing critical analysis, so the authors must include the limitations and individual variations of the subjects.

In results (3.1) a baseline table and subject characteristics must be done, and co-morbidities must be informed (also, BMI)

Figure 1… The il-6 levels graph was missing

In all the Figures, the description (legend) has to be more accurate ( N° subjects, statistic treatment, complete description. etc). The figures are small and a little messy, please improve them.

Figure 3, why not analyze normal/overweight? In that context, the baseline was of the “total subjects” or only with the subjects that presented elevated levels in Figure 1?

Maybe a survival graph should be more informative about the MCP-1 and the risk factor observed.

Why the authors didn’t measure TNF-a?

In the discussion, a deeper analysis of the EPA+DHA must be done (more molecular description)

Ref 51, and PPARalpha?

Line 296-297 ???????

Author Response

We sincerely thank the Editor for the interest in our patient cohort and results. Following your suggestions, we re-evaluated our manuscript and focused on the comments of the Reviewers. We sincerely thank the Reviewers for their constructive comments, which we found very helpful towards improving the quality of our study. Accordingly, specific changes have been made in the manuscript, based on these comments, as it is described in detail below in a point-by-point description of the changes introduced, and on how Reviewer’s concerns were addressed. Changes within the manuscript are indicated in red.

Reviewer: Interesting work, but the manuscript should be improved: Nutritional support and Circulating cytokines in patients with heart failure. Is it possible to change the clinical course of these patients? The title opens a question that is expected to be resolved in the text, but I am not sure if they answer it. Maybe the title should be changed.

Authors: as the reviewer suggested, the title was changed

Reviewer: Line 21... the phrase is in a different letter typo, same in line 45 (and lines 6, 59, 141,142), line 30 C-RP? Maybe is CRP

Authors: these typo´s have been corrected

Reviewer: In methodology, authors should include the n-3 (EPA, DHA) and omega-6 (LA?) concentration in every dose administrated.

Authors: this information was included in the materials and methods section

Reviewer: 19 patients is a low average for doing critical analysis, so the authors must include the limitations and individual variations of the subjects. 

Authors: we agree with the reviewer regarding this point. This point was discussed deeply in the revised version of our manuscript.

Reviewer: In results (3.1) a baseline table and subject characteristics must be done, and co-morbidities must be informed (also, BMI)

Authors: Tables were attached in another document, which seemed not to be visible to the reviewers. We apologize about this matter; tables were included in the revised version of our manuscript.

Reviewer: Figure 1… The il-6 levels graph was missing

Authors: Figure 1 only represents significant associations. For avoiding confusion, this has been explained in the revised version of our manuscript.

Reviewer: In all the Figures, the description (legend) has to be more accurate ( N° subjects, statistic treatment, complete description. etc). The figures are small and a little messy, please improve them.

Authors: Figures and legends have been edited as suggested by the reviewer

Reviewer: Figure 3, why not analyze normal/overweight? In that context, the baseline was of the “total subjects” or only with the subjects that presented elevated levels in Figure 1? 

Authors: Figure 3 represents changes in all patients. This information has been included in the results section and in the figure legend.

Reviewer: Maybe a survival graph should be more informative about the MCP-1 and the risk factor observed.

Authors: we do not have the exact date of dead, only a report of survival, for that reason it was not possible to make a survival analysis. Table 3 with all the multivariate analysis was included

Reviewer: Why the authors didn’t measure TNF-a?

Authors: the measured cytokines were selected based on their novelty and literature reports in HF. For future studies we can include its measurement.

Reviewer: In the discussion, a deeper analysis of the EPA+DHA must be done (more molecular description)

Authors: This information has been included in the revised version of our manuscript.

Reviewer 2 Report

Comments and Suggestions for Authors

In Herrera-Martinez’s paper the investigators aimed to evaluate circulating cytokines levels in a cohort of patients with HF after receiving nutritional support with Mediterranean diet alone or in combination with a hypercaloric and hyperproteic oral nutritional supplement. Circulating cytokines were determined at baseline and after 24 weeks of nutritional support in a cohort of patients that were previously included in an open label and controlled clinical study.

The paper is well written and organized logically in every part, characteristics that make it easy readable and understandable. The text is consistent with the information presented in the figures which are clear and correct. The results are comprehensively and clearly reported. The statistical analyses are appropriate and most of the bibliographic references derive from journals with high impact factors.

Here are some specific comments:

ABSTRACT:

 The abstract is concise and informative and matches the content of the paper. The authors briefly but clearly mention the background, the aim, materials and methods, key results and conclusions of the study

INTRODUCTION:

 The introduction provides the right and useful background information on the research topic. The authors clearly explain motivation and present the aim of the study in the context of what is already known about the topic.

1 minor comment:

Row 57: Ref 11. Please check this reference since it seems difficult to find a correspondence between the sentence and the reference content.

MATERIALS AND METHODS:

 1 minor comment:

They were included patients who have been hospitalized for HF in the previous 12 months (as specified on CT.Gov), or patients who have been hospitalized in the previous 6 months (as described in the paper)? In case of the latter, please specify it in the paper and the reason for such a change.

 Please consider to add the primary outcomes for study NCT05848960 (row 88).

RESULTS

1 Major Comment:

All tables are missing

Minor comments:

Row 183 Typo 2B instead of 2A

DISCUSSION :

This section is well organized and easy to understand. Moreover it fairly analyzes data, describes the meaning of the results and evaluates the strength points but also the limitations of the work.

2 Minor comments:

- Row 296: please consider to add the quoted article (Adipocyte 2022 etc) to bibliographic references list and delete it from the text

- Please consider to add on row 229 (at the end of the first sentence on HF prevalence) the bibliographic reference.

Comments on the Quality of English Language

The English style is fine and requires just minor checks.

Author Response

We sincerely thank the Editor for the interest in our patient cohort and results. Following your suggestions, we re-evaluated our manuscript and focused on the comments of the Reviewers. We sincerely thank the Reviewers for their constructive comments, which we found very helpful towards improving the quality of our study. Accordingly, specific changes have been made in the manuscript, based on these comments, as it is described in detail below in a point-by-point description of the changes introduced, and on how Reviewer’s concerns were addressed. Changes within the manuscript are indicated in red.

Reviewer: In Herrera-Martinez’s paper the investigators aimed to evaluate circulating cytokines levels in a cohort of patients with HF after receiving nutritional support with Mediterranean diet alone or in combination with a hypercaloric and hyperproteic oral nutritional supplement. Circulating cytokines were determined at baseline and after 24 weeks of nutritional support in a cohort of patients that were previously included in an open label and controlled clinical study.

The paper is well written and organized logically in every part, characteristics that make it easy readable and understandable. The text is consistent with the information presented in the figures which are clear and correct. The results are comprehensively and clearly reported. The statistical analyses are appropriate and most of the bibliographic references derive from journals with high impact factors.

Here are some specific comments:

ABSTRACT:

  • The abstract is concise and informative and matches the content of the paper. The authors briefly but clearly mention the background, the aim, materials and methods, key results and conclusions of the study

INTRODUCTION:

  • The introduction provides the right and useful background information on the research topic. The authors clearly explain motivation and present the aim of the study in the context of what is already known about the topic.

Reviewer: Row 57: Ref 11. Please check this reference since it seems difficult to find a correspondence between the sentence and the reference content.

 Authors: This reference has been corrected. We thank the reviewer for noticing this out.

MATERIALS AND METHODS:

Reviewer: They were included patients who have been hospitalized for HF in the previous 12 months (as specified on CT.Gov), or patients who have been hospitalized in the previous 6 months (as described in the paper)? In case of the latter, please specify it in the paper and the reason for such a change.

Authors: The study included patients who have been hospitalized for HF in the previous 12 months, this mistake was corrected in the revised version or our manuscript.

Reviewer: Please consider to add the primary outcomes for study NCT05848960 (row 88).

Authors: This information was included in the revised version of our manuscript.

RESULTS

Reviewer: All tables are missing

Authors: Tables were attached in another document, which seemed not to be visible to the reviewers. We apologize about this matter; tables were included in the revised version of our manuscript.

Reviewer: Row 183 Typo 2B instead of 2A

Authors: This typo was corrected; we thank the reviewer for noticing this out.

DISCUSSION :

This section is well organized and easy to understand. Moreover it fairly analyzes data, describes the meaning of the results and evaluates the strength points but also the limitations of the work.

Reviewer:  Row 296: please consider to add the quoted article (Adipocyte 2022 etc) to bibliographic references list and delete it from the text

Authors:  This reference has been added

Reviewer:  Please consider to add on row 229 (at the end of the first sentence on HF prevalence) the bibliographic reference.

Authors: This reference has been added

Reviewer 3 Report

Comments and Suggestions for Authors

Dear authors,

I have carefully studied the manuscript entitled “Nutritional Support and Circulating Cytokines in Patients with Heart Failure. Is It Possible to Change the Clinical Course of These Patients?” by Aura D. Herrera-Martinez et al. The manuscript is very interesting but there are some changes that need to be made:

1.     Lines 16-18. The sentences do not have a cursivity. Please rewrite.

2.     Line 20. Please make the text more fluent (e.g. The aim of our study was to…).

3.     The abstract should include (1)Background, (2)Methods, (3)Results, (4)Discussion, (5)Conclusion. Please follow the template from the mdpi site.

4.     Lines 47-51. Please rephrase, the sentence is too long and hard to follow.

5.     Line 66. Please add examples of types of food that increase the inflammation.

6.     Lines 22, 45, 46, 59, 141, and 142 - The font used for the text is different from the font used for the whole article. Please correct this inconsistency.

7.     I can not find tables 1 and 2 that are described in the article.

8.     In the discussion section, please compare your results with the results in the literature.

9.     It would be useful for readers to add a separate chapter discussing the purpose of the study.

10.  It would be better to have a dedicated chapter that discusses the limitations.

Comments on the Quality of English Language

Minor English editing is required

Author Response

We sincerely thank the Editor for the interest in our patient cohort and results. Following your suggestions, we re-evaluated our manuscript and focused on the comments of the Reviewers. We sincerely thank the Reviewers for their constructive comments, which we found very helpful towards improving the quality of our study. Accordingly, specific changes have been made in the manuscript, based on these comments, as it is described in detail below in a point-by-point description of the changes introduced, and on how Reviewer’s concerns were addressed. Changes within the manuscript are indicated in red.

Reviewer: Dear authors,

I have carefully studied the manuscript entitled “Nutritional Support and Circulating Cytokines in Patients with Heart Failure. Is It Possible to Change the Clinical Course of These Patients?” by Aura D. Herrera-Martinez et al. The manuscript is very interesting but there are some changes that need to be made:

  1. Reviewer:Lines 16-18. The sentences do not have a cursivity. Please rewrite.

Authors: This typo has been corrected

  1. Reviewer:Line 20. Please make the text more fluent (e.g. The aim of our study was to…).

Authors: this change was performed

  1. Reviewer: The abstract should include (1)Background, (2)Methods, (3)Results, (4)Discussion, (5)Conclusion. Please follow the template from the mdpi site.

Authors: as suggested by the reviewer, template was followed

  1. Reviewer:Lines 47-51. Please rephrase, the sentence is too long and hard to follow.

Authors: as suggested by the reviewer, the paragraph was rephrased

  1. Reviewer: Line 66. Please add examples of types of food that increase the inflammation.

Authors: This information was added

  1. Reviewer:Lines 22, 45, 46, 59, 141, and 142 - The font used for the text is different from the font used for the whole article. Please correct this inconsistency.
  2. Reviewer: I can not find tables 1 and 2 that are described in the article.

Authors: Tables were attached in another document, which seemed not to be visible to the reviewers. We apologize about this matter; tables were included in the revised version of our manuscript.

  1. Reviewer:In the discussion section, please compare your results with the results in the literature.

Authors: Study results have been compared with others described in the literature.

  1. Reviewer:It would be useful for readers to add a separate chapter discussing the purpose of the study.

Authors: as suggested by the reviewer, this section was added

  1. Reviewer:It would be better to have a dedicated chapter that discusses the limitations.

Authors: Limitations have been enlarged in the revised version of our manuscript and a new chapter was added.

Reviewer 4 Report

Comments and Suggestions for Authors

Dear authors,

This study was conducted to nutritional support and circulating cytokines in patients with heart failure. Basically, this manuscript is well-written and the study has a good rationale and sound hypotheses. However, it needs more revisions. Some suggestions are recommended for the authors’ consideration.

Abstract

(1) Please do not express p<0.05, but express the exact p-value (p=0.XXX) in each all variable.

(2) The journal instructs that the abstract should not exceed 200 words and should be unstructured. Please note that you should to make significant changes to the abstract so that it did not exceed the 200-word limit the journal prescribes.

Introduction

(1) The authors should more explain the concept of increased circulating levels of cytokine. Besides, the research gap between nutritional support and circulating cytokines in the article is not clear, and the authors should strengthen the gap illustration according to the prior research.

(2) Re-format the entire manuscript according to the journal's guidelines. For example, Line 44 in Introduction, “hypertension, atrial fibrillation and sarcopenia (1, 2),” à hypertension, atrial

fibrillation and sarcopenia [1,2].

Materials and methods:

Well-written.

Results

I am not able to find Table 1 and 2. Please add Tables in manuscript.

Line 155: Abbreviations should be defined in the first instance. Please adjust whole manuscript.

For example, BMI à body mass index (BMI)

Please revise all results to two decimal places from one or three decimal places in whole manuscript.

Discussion

Please add several sentences of (1) application in this field and (2) more limitations of this study.

Comments on the Quality of English Language

I recommend that this manuscript have to be edited by an English professional editor for more readable. There are many typo and grammatical errors.

Author Response

We sincerely thank the Editor for the interest in our patient cohort and results. Following your suggestions, we re-evaluated our manuscript and focused on the comments of the Reviewers. We sincerely thank the Reviewers for their constructive comments, which we found very helpful towards improving the quality of our study. Accordingly, specific changes have been made in the manuscript, based on these comments, as it is described in detail below in a point-by-point description of the changes introduced, and on how Reviewer’s concerns were addressed. Changes within the manuscript are indicated in red.

Reviewer: Abstract (1) Please do not express p<0.05, but express the exact p-value (p=0.XXX) in each all variable.

 Authors: as suggested by the reviewer, the exact value was used.

Reviewer:  (2) The journal instructs that the abstract should not exceed 200 words and should be unstructured. Please note that you should to make significant changes to the abstract so that it did not exceed the 200-word limit the journal prescribes.

Authors: as suggested by the reviewer, the abstract was edited.

Reviewer:  Introduction (1) The authors should more explain the concept of increased circulating levels of cytokine. Besides, the research gap between nutritional support and circulating cytokines in the article is not clear, and the authors should strengthen the gap illustration according to the prior research.

 Authors: this relation was included in the introduction and discussion sections of the revised manuscript.

Reviewer:  (2) Re-format the entire manuscript according to the journal's guidelines. For example, Line 44 in Introduction, “hypertension, atrial fibrillation and sarcopenia (1, 2),” à hypertension, atrial fibrillation and sarcopenia [1,2].

 Authors: as suggested by the reviewer, the manuscript was edited

Reviewer: Results. I am not able to find Table 1 and 2. Please add Tables in manuscript.

 Authors: Tables were attached in another document, which seemed not to be visible to the reviewers. We apologize about this matter; tables were included in the revised version of our manuscript.

Reviewer: Line 155: Abbreviations should be defined in the first instance. Please adjust whole manuscript. For example, BMI à body mass index (BMI)

 Authors: Abbreviations have been revised in the new version of the manuscript

Reviewer: Please revise all results to two decimal places from one or three decimal places in whole manuscript.

 Authors: Decimals have been revised in the new version of the manuscript

Reviewer: Discussion. Please add several sentences of (1) application in this field and (2) more limitations of this study.

 Authors: as suggested by the reviewer, discussion was edited and limitations were enlarged and presented in a separate chapter.

Round 2

Reviewer 3 Report

Comments and Suggestions for Authors

The authors significantly improved the quality of the article. It can now be published